# Sensing and Analysis of Greenhouse Gas Emissions from Rice Fields to the Near Field Atmosphere

**DOI:** 10.3390/s22114141

**Published:** 2022-05-30

**Authors:** Panneerselvam Rajasekar, James Arputha Vijaya Selvi

**Affiliations:** Kings College of Engineering, Affiliated to Anna University, Punalkulam 613303, Tamil Nadu, India; randdece@gmail.com

**Keywords:** greenhouse gases (GHGs), irrigation treatments, automatic gas chamber, rice fields

## Abstract

Greenhouse gas (GHG) emissions from rice fields have huge effects on climate change. Low-cost systems and management practices to quantify and reduce GHGs emission rates are needed to achieve a better climate. The typical GHGs estimation processes are expensive and mainly depend on high-cost laboratory equipment. This study introduces a low-cost sensor-based GHG sampling and estimation system for rice fields. For this, a fully automatic gas chamber with a sensor-integrated gas accumulator and quantifier unit was designed and implemented to study its performance in the estimation efficiency of greenhouse gases (CH_4_, N_2_O, and CO_2_) from rice fields for two crop seasons. For each crop season, three paddy plots were prepared at the experimental site and then subjected to different irrigation methods (continuous flooding (CF), intermittent flooding (IF), and controlled intermittent flooding (CIF)) and fertilizer treatments to study the production and emission rates of GHGs throughout the crop growing season at regular intervals. A weather station was installed on the site to record the seasonal temperature and rainfall events. The seasonal total CH_4_ emission was affected by the effects of irrigation treatments. The mean CH_4_ emission in the CIF field was smaller than in other treatments. CH_4_ and N_2_O emission peaks were high during the vegetative and reproductive phases of rice growth, respectively. The results indicated that CIF treatment is most suitable in terms of rice productivity and higher water use efficiency. The application of nitrogen fertilizers produced some peaks in N_2_O emissions. On the whole, the proposed low-cost GHGs estimation system performed well during both crop seasons and it was found that the adaption of CIF treatment in rice fields could significantly reduce GHG emissions and increase rice productivity. The research results also suggested some mitigation strategies that could reduce the production of GHGs from rice fields.

## 1. Introduction

Rice is the second major staple food in 48 countries of Asia and is being grown on about 153 Mha, which is equal to 11% of the world’s arable land [1]. A major concern in the cultivation of rice is the fact that it is one of the main agricultural sources of emissions of greenhouse gases (GHGs) like CH_4_, N_2_O, and CO_2_ [2]. Rice crops have 169% higher global warming potential than maize and 460% higher than wheat crops. The Global Warming Potential (GWP) rate is higher in southern and eastern parts of India due to the larger extension of rice cultivation fields. This is mainly due to the contribution of higher CH_4_ emission to total greenhouse gas emissions in rice production compared to othercrops. It is estimated that by 2030, the demand for rice production may increase by 40% due to the growing population. Hence, the resulting GHG emission rate may increase by 35% to 60% [3], which will lead to increased global warming [4]. The excessive application of inorganic fertilizers in the rice field to increase productivity may also increase GHGs emissions from rice fields. GHG emissions from paddy fields depend on various factors like the irrigation-water level, amount of fertilizer used, rice varieties, and soil parameters [5]. Increasing the soil temperature and moisture content of paddy fields increases the N_2_O and CH_4_ emissions by affecting the methanogenesis process, aerobic and anaerobic processes, respectively [6,7]. A higher carbon cycling rate is found in rice fields due to the flooding nature of irrigation and excess application of fertilizers and composts [8,9,10]. The release of CH_4_ from the rice field happens by the diffusion loss of dissolved methane and evaporation losses [11,12]. In aerobic and anaerobic soil conditions, N_2_O is produced by the microbial processes of nitrification and denitrification. N_2_O is mainly emitted after strong rainfall events and the application of fertilizers (mainly N fertilizer) onto the rice fields [13]. The production and emission of GHGs fluxes are depicted in Figure 1 and Figure 2.

The future rice cropping system should focus on achieving increased rice productivity and decreased GHGs emissions [14,15]. Several strategies are being implemented for increasing the productivity of rice field, however, the resulting environmental and field-level impacts on climate change have not been elaborated in detail. By accessing various academic portals, it is found that there is a lack of information on how rice fields affect GHG emissions and estimations for a longer period [16]. To understand the spatial and temporal variability of GHG fluxes, long-term continuous datasets are needed. There are two methods for quantifying the GHG emissions from soil, namely the micrometeorological method and the chamber-based method. Micrometeorological measurements require costly equipment and estimation of each GHG is difficult due to the involvement of various soil and environment factors. Chamber-based methods may be useful due to their low cost nature and ability to measure GHGs from soil to atmosphere without requiring an additional power supply. Hence, the introduction and implementation of a low-cost sensor-based automatic gas sampling and estimation method are needed to replace the existing gas sampling method and increase the efficiency in estimating the GHGs fluxes from paddy fields.

## 2. Background and Related Works

From the literature survey, several methods exist to measure GHG emissions from the soil surface to the atmosphere. For this, micrometeorological and chamber-based estimation methods are widely used. In [17,18] the authors have developed different static gas chambers with various materials like plastic, metal, and glass to sample greenhouse gases.Mostly, syringes are used to collect the sampled gases from gas chambers to calculate the concentration of each gas flux in the laboratory.The authors in [19] utilized a static gas chamber and the gas chromatography technique to analyze GHGconcentrations. However, the post-processes involved in these techniques are very tedious and costly due to their dependency on laboratory-based instruments and also they are labor and time-consuming. The authors in [20] have developed plastic gas chambers to sample GHG fluxes. Here, more attention was focused on choosing the best type of plastic material to design gas chambers because, some plastic materials are permeable to GHGs and/or they also can react with GHGs and produce other gases like CO, NO_3_, etc. The authors in [21] addressed the difficulties and challenges associated with installing gas chambers in rice fields and they also investigated the lifetime of these gas chambers. The authors of [22] have developed a CH_4_ measurement system in which the collection of gas samples was done on a weekly basis. Due to the large time interval between each gas sample, this system could not monitor GHGs dynamics thoroughly. The authors in [23] designed and implemented an automated gas chamber which samples gases into test-tubes at regular intervals but still they need laboratory-based measurements. Estimation of the magnitude of GHG fluxes from the environment in real-time has begun recently [24]. Very few long-term datasets on GHGs emissions for rice fields are available due to the challenges involved in designing a continuous multi-gas estimation system. Generally, the soil GHG fluxes are measured using static chambers that are manually closed for a fixed period to collect gas samples [25]. Also, the short time variations in GHG fluxes are not being detected by these static chamber-based methods. This leads to a lost opportunity to thoroughly understand GHG production and emissions and their impacts on climate change. Also, there is no autonomous system for detecting and estimating GHGs emissions in rice fields throughout the crop growing period. All the existing methodologies from the review use manual collection of gases using fixed gas chambers and laboratory-based gas sampling techniques which leads to time delays [26]. There are very few works that have replaced conventional gas sampling methods with sensor-based gas sampling and estimation methods [27]. Therefore, the main research contributions reported in this paper are as follows:(1)Rice field’s productivity using different irrigation and fertilizer treatments and their impacts on climate change is investigated.(2)A low cost sensor-based automated in-situ GHGs measuremnet system an an analysis of seasonal (wet and dry season)variations in productivity and GHG emissions is developed and implemented.(3)Suitable irrigation and fertilizer treatment towards achieving higher productivity with reduced GHG emissions are suggested.

## 3. Experimental Setup

Gas fluxes from rice fields can be measured by various approaches like points scale chambers, micrometeorological techniques, gas chromatography, spectroscopic methods and optical gas imaging. All the measurement techniques need very expensive instruments and involve complex operations. Hence, in this study we designed a cost-effective automated gas sampling and estimation system as shown in Figure 3. For this purpose, a lightweight and breakage-resistant automated gas chamber for the collection of GHG fluxes was designed [28]. The height, breadth and width of the designed gas chamber were 310 cm, 60 cm and 90 cm, respectively, as shown in Figure 4. The top of the chamber was kept higher than the rice plant height to avoid disturbances to the rice growth. The chamber shape and area were decided by the method of rice plant transplantation. Before sampling soil gas, the chamber was inserted into the soil at approximately 30 cm depth in each irrigation treatment field. A wooden gas sampling box (GSB) with the size of 60 cm × 30 cm × 30 cm (length × breadth × height) was prepared to hold all the components required for the gas sampling and estimation process as shown in Figure 5. A gas accumulator and quantifier unit (GAQU) integrated with GHG sensors (CH_4_ sensors: TGS 2611 and MQ4), N_2_O sensor (MICS-4514), and CO_2_ sensor (MQ135) with a temperature (DHT22) and pressure sensor) was developed and placed in the gas sampling box. Four solenoid valves and two air pumps were utilized for bringing gases in and out of GAQU. A programmed electronic controller module for sampling and quantifying the gases was inserted into GSB with a rechargeable battery. A raised steel bed to hold the GSB was designed and installed in the rice field. The meteorological data (temperature, rainfall, wind speed, and direction) were collected by a weather meter placed over the raised steel bed.

## 4. Experimental Methodologies

The future rice cropping system should focus on combining increased rice productivity and decreased GHGs emissions. Hence, this research work focused on investigating the effects of different irrigation and fertilizer treatments on greenhouse gas emissions and rice productivity in rice fields. For this, two crop seasons were chosen. For each crop season, three paddy plots were prepared and subjected to different irrigation (continuous flooding (CF), intermittent flooding (IF), and controlled intermittent flooding (CIF)) and fertilizer treatments. A fully automated gas chamber was designed and installed in each paddy field. A gas accumulation and quantifying unit (GAQU) equipped with low-cost GHG sensors was constructed. The gas flux data was collected from the automatic gas chambers and stored in the GAQU and the concentration of each gas flux was estimated.

The research site was situated in the Thanjavur district of Tamil Nadu state (India) (10.712625, 79.149450), some 7 km away from the town of Thanjavur. Two crop growing seasons were chosen (wet season and dry season) to implement and investigate the objectives of the proposed research. The wet season was from October 2020 to January 2021 and the dry season was from January 2021 to May 2021. The average temperature, rainfall amount, and wind speed of the experimental site were recorded. Three experimental paddy sites with an area of 12 m × 12 m were prepared to test the proposed research objectives. In both seasons, rice (IR36 variety) was transplanted into all three rice fields. All the paddy sites were irrigated by a shallow tube well located nearby the site. Three irrigation treatments with uniform fertilizer application were followed in all three proposed rice fields. They are: (1) continuous flooding (CF), (2) intermittent flooding (IF), and (3) controlled intermittent flooding (CIF). For CF, the rice field was irrigated to 30 cm water from the soil surface and this was maintained throughout the crop growing season. In the IF site, the irrigation water was maintained to 30 cm every 3 days whereas, in the CIF site, the rice field was irrigated according to the soil moisture prediction and irrigation scheduling method. The first fertilizer application (FA-1) was done with N fertilizer (urea) and diammonium phosphate (DAP) in all three rice fields after transplanting. Twenty days after transplanting, a second fertilizer application (FA-2) was done with 60 kg/ha of urea and 50 days after transplanting, a third fertilizer application (FA-3) was done with 120 kg/ha of P_2_O_5_ and 60 kg/ha of urea as listed in Table 1. The herbicide and pesticide were applied in all three fields whenever required. The installation of gas chambers, GSB and weather meter in the rice fields, the gas sampling and estimation process started for both wet and dry seasons. Three processes were utilized to quantify GHGs fluxes in this research as depicted in Figure 6. They are gas accumulation (GA), gas estimation (GE), and gas removal (GR). In the GA process, the gas fluxes formed in each gas chamber were captured and accumulated in the GAQU by activating the corresponding solenoid valve and air pump for each chamber. The activation time of the solenoid valve and air pump was fixed by the threshold value of the pressure sensor installed in the GAQU. In this work, this was approximately 75 s. After the gas accumulation process of all chambers, the GE process starts where all the solenoid valves and air pumps utilized were turned off and the GHGs estimation process started after a rest period of 30 s. The GHGs concentrations were obtained using the low-cost sensors integrated with GAQU during the estimation process. The timing of the GE process was fixed to 20 min for each chamber. The raw values of sensors were converted into gas concentrations using the sensor manufacturers’ calibration datasheets. The raw values of each sensor are converted into concentrations (ppm) using the following formulas,
V0 = Raw Sensor Value × 5/1023 (1)
RS = (5 − V0) × 1000/V0 (2)
Gas in PPM = pow(RS/R0, −2.95) × 1000 (3)
where the RS is sensor resistance at target, R0 is the sensor resistance in clean air and V0 is the sensor voltage converted from raw value. The GHG emissions were estimated by the following equation:
J = (dc/dt) × (M/Vo) × (P/Po) × (To/T) × H(4)
where J stands for the GHG (CO_2_, CH_4_ and N_2_O) emission in mg m^−2^h^−1^, M stands for the molar mass of each GHG, P stand for pressure, t is the temperature, H is the chamber height, and To, Po and Vo are the temperature, pressure and volume, respectively.

The GR process was started after the gas accumulation and estimation process. The gases in the GAQU were removed/released by the second air pump installed in the GSB. The timing of the GR process was fixed to 60 s. Three rice fields in both wet and dry seasons under CF, IF and CIF irrigation treatments were tested using the proposed GHGs sampling and estimation method. The GA, GE, and GR processes were regularly repeated until the rice harvest. The water use efficiency, rice productivity, and GHGs rate of data were acquired for all three rice fields. Dry root weight, tiller number, leaf area index, leaf number, and rice grain weight for each field were calculated and compared.

## 5. Results and Data Analysis

The cumulative soil moisture content in all rice fields due to the irrigation and rainfall events were calculated to find the irrigation water use efficiency against rice productivity and it was found larger for CIF treatment than the CF and IF treatments in both the wet and dry seasons. The CIF site attained higher irrigation water savings than other irrigation methods. In addition, due to the greater rainfall amount in the wet season, the total water use was significantly higher. The amount of rainfall and the average temperature of the air in the crop growing area were recorded and found distinctly different between wet and dry seasons as plotted in Figure 7.

Figure 8 illustrates the seasonal emissions of GHGs from the rice fields in both seasons. The seasonal magnitude of CH_4_ gas emission was found to differ in both seasons. The CH_4_ gas flux was large during the yearly growth stage of the rice plants. The total CH_4_ emissions in the wet season were higher than in the dry season. The effect of CIF treatment on CH_4_ emissions was found to be smaller than that of other treatments. The N_2_O fluxes were found to be sporadic in both seasons, regardless of the irrigation treatment. Large N_2_O emission peaks were observed after N fertilizer application. In the dry season, the N_2_O emission was three times greater than during the wet season. The N_2_O emission from CIF fields was found higher than with other irrigation treatments. The increased soil respiration rate in the CIF field resulted in increased production of N_2_Oand CO_2_ emissions during both seasons. On the other hand, CO_2_ emissions from all three rice fields were high whenever the soil became drier. In Figure 9 it was clearly seen that the CO_2_ emissions in the CIF field were comparatively higher than in the other fields. Since, there was no significant seasonal difference in the CO_2_ emission rate, we have only depicted the CO_2_ variations of the three irrigation regimes for the dry season. The irrigation was discontinued in all fields 15 days before harvesting. To investigate the effects of irrigation treatments, 15 rice plants per site were randomly chosen, and plant properties (panicle length, plant height, and roots density with length and weight of 1000 seeds) were manually calculated and are listed in Table 2. The effect of crop season (wet or dry) on the grain yield was significant. Due to the higher solar radiation during the dry season, the yield was greater than in the wet season. The controlled irrigation treatment resulted in higher water use efficiency than the other treatments.

The effect of crop season played a vital role in the GHG emissions. The wet season resulted in greater CH_4_ emissions than the dry season. During the dry season, a significant difference in CH_4_ and N_2_O emissions was found between the CF and CIF treatments. The higher temperature after the N application resulted in increasing N_2_O emissions due to the dry conditions. Under flooded soil conditions N_2_O fluxes were found to be reduced after top-dressing. Emission of GHGs from the rice fields, especially N_2_O and CH_4_, can be reduced by incorporating various management practices like irrigation pattern adjustments, management of organic additives, use of appropriate N fertilizer rates, suitable tillage practices, cropping regimes, selection of suitable cultivars, use of cover crops and nitrification inhibitors [29]. Irrigation pattern is the great influencing factor for GHG emissions. Soil moisture dynamics due to irrigation regimes mainly affect the oil redox potential which is the source for regulating the consumption and rate of release of GHGs. Hence, adapting controlled intermittent flooding could result in CH_4_emission reduction due to the time intervals that cause a switch from aerobic to anaerobic soil conditions [30]. CIF treatment resulted in lower GHGs emissions and higher rice productivity due to the less application of irrigation water to the field. The applied nitrogen fertilizer is not being utilized completely by rice crops [31]. Hence, the appropriate fertilizer application strategies such as application rate modification, exact application time, usage of slow-release fertilizers, avoidance of over-application, and precise placement of fertilizer into the soil can have a substantial impact for reducing GHGs emissions. Reduced quantities of N fertilizer in the soil can lower the N_2_O emissions [32,33]. Sulphate fertilizer application is a suitable option for increasing alternative electron acceptors in the soil to reduce CH_4_ emissions. Nitrification inhibitors can be used to reduce CH_4_ and N_2_O emissions by delaying the nitrification process and reducing the availability of NO_3_ for the de-nitrification process [34,35].

## 6. Conclusions

This research work developed and implemented a low-cost sensor-based automated in-situ GHG measurement system. The proposed system was utilized as a test bed to investigate rice field productivity and the impacts on climate by measuring seasonal GHG emissions from rice fields under different irrigation schemes (CF, IF and CIF) and fertilizer treatments. The results obtained from this work demonstrate the reliability of the low cost sensor-based GHG measurement system to measure GHG emissions from rice fields to the atmosphere.

CIF treatment reduced CH_4_ emissions compared to the IF and CF treatments and did not have much influence on N_2_O emissions. The water use was considerably reduced by CIF treatment and the resulting rice grain yield was higher than with other irrigation treatments. The application of nitrogen fertilizers resulted in some N_2_O peaks and lengthy a drying period resulted in higher CO_2_ emissions. On the whole, the proposed low-cost GHGs estimation system performed well during both crop seasons and it was found that the adoption of CIF treatment in the rice fields could significantly reduce the GHGs emission and increase rice productivity.

Generally, farmers won’t care about climate change and environmental management activities if they are not reflected in their income. Hence, the adoption of CIF irrigation treatment with essential mitigation strategies into the rice fields of local farmers would achieve increased rice productivity with an acceptable reduced GHG emission rate. The performances of automated gas chambers installed in rice fields were accurate until harvesting in both crop seasons. The strategies used to sample gases in this work may be useful for increasing the accuracy of measurements and durability of sensors. The proposed GHG estimation system after some further customization could also be used to estimate the GHGs emitted by industries, transportation, agriculture and other commercial and residential areas.

## Figures and Tables

**Figure 1 sensors-22-04141-f001:**
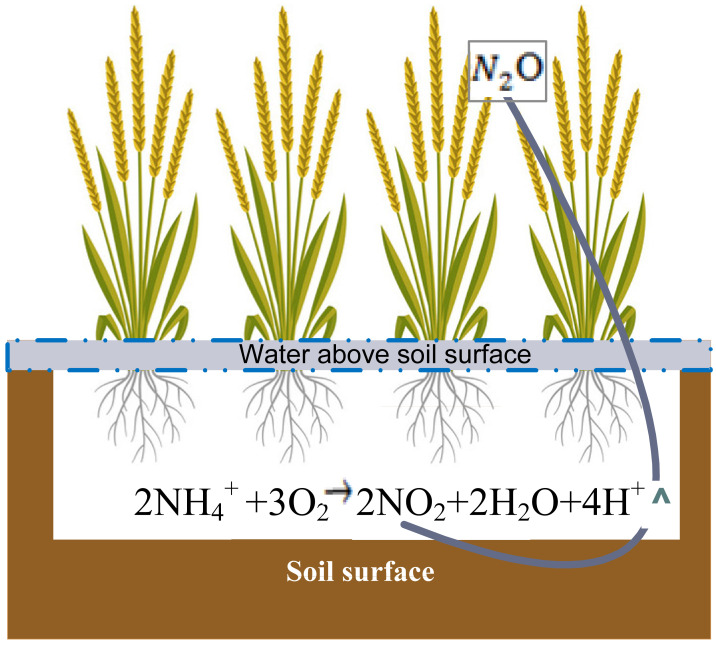
The process of N_2_O production and emission from rice field.

**Figure 2 sensors-22-04141-f002:**
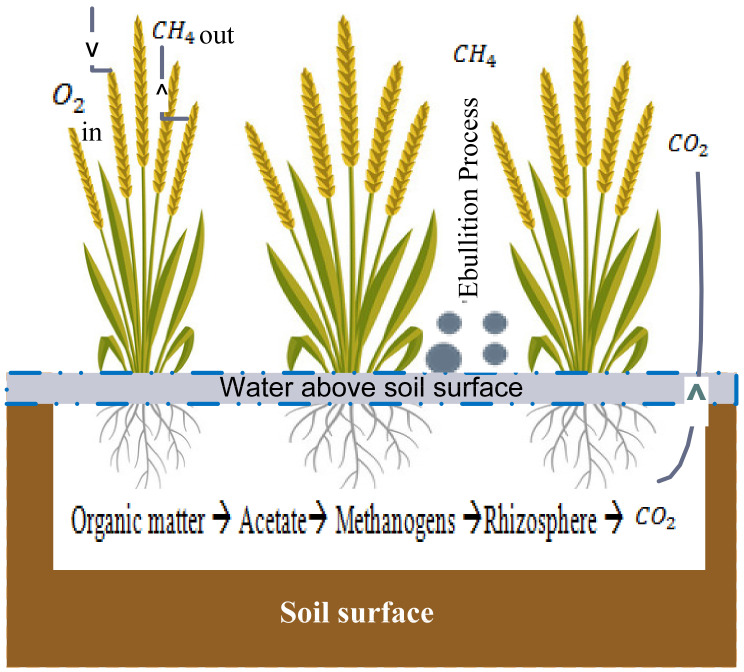
The process of CH_4_and CO_2_ production and emission from rice field.

**Figure 3 sensors-22-04141-f003:**
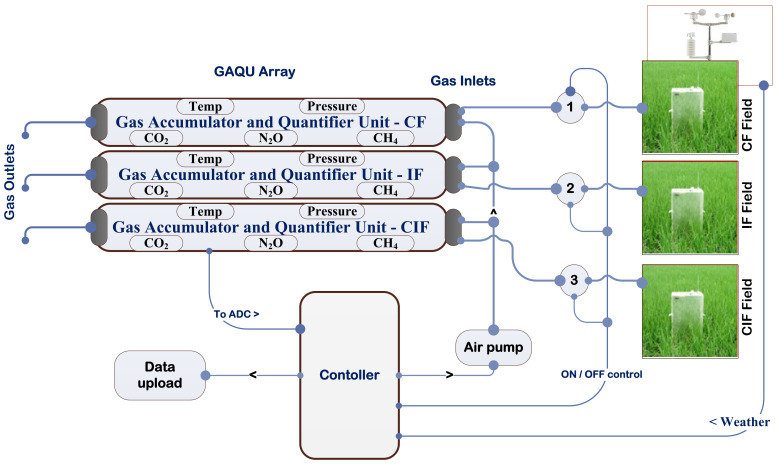
Automated sensor assisted gas sampling and estimation system.

**Figure 4 sensors-22-04141-f004:**
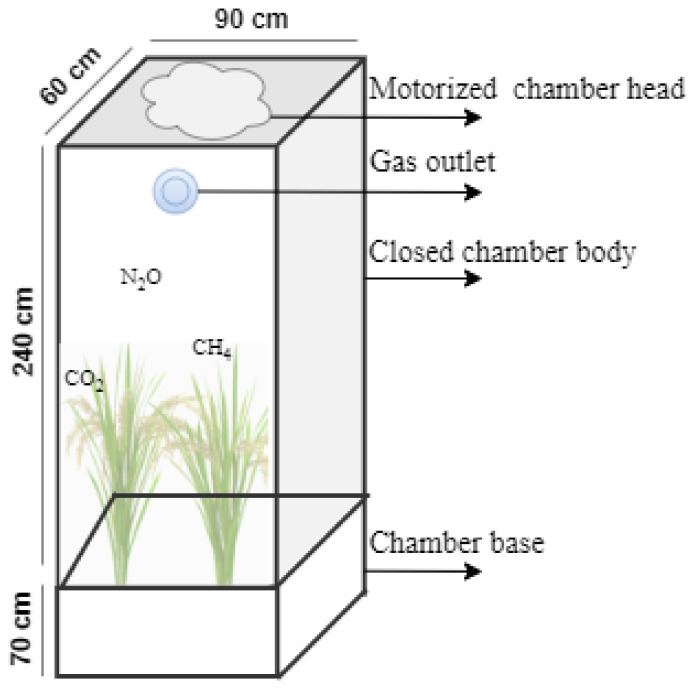
Proposed gas chamber design.

**Figure 5 sensors-22-04141-f005:**
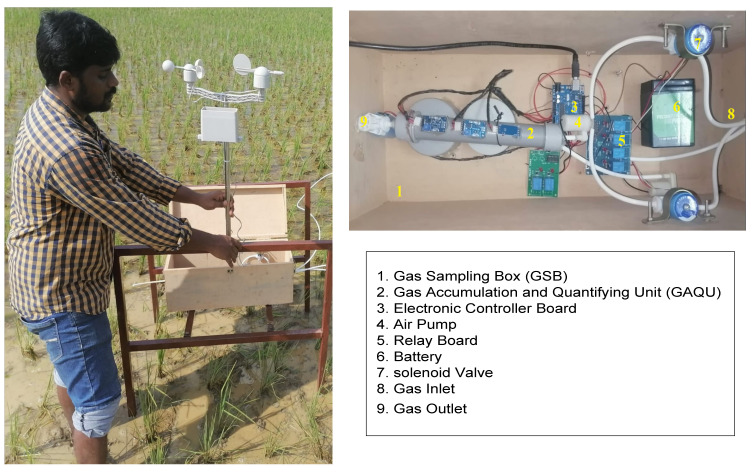
Photocopy of developed Gas sampling box with GAQU installed in rice field.

**Figure 6 sensors-22-04141-f006:**
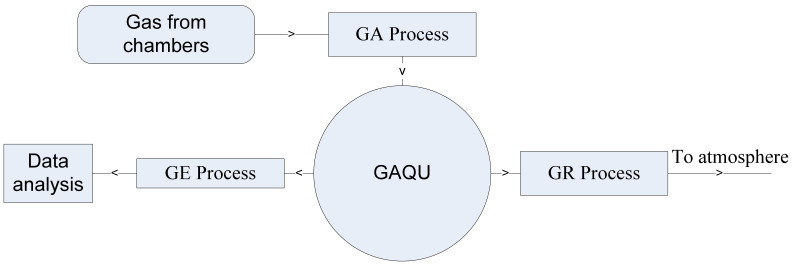
Three processes utilized for GHG estimation.

**Figure 7 sensors-22-04141-f007:**
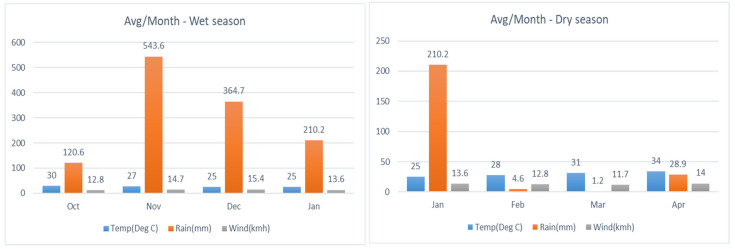
Average monthly temperature, rainfall and wind speed of the experimental site in both wet and dry season.

**Figure 8 sensors-22-04141-f008:**
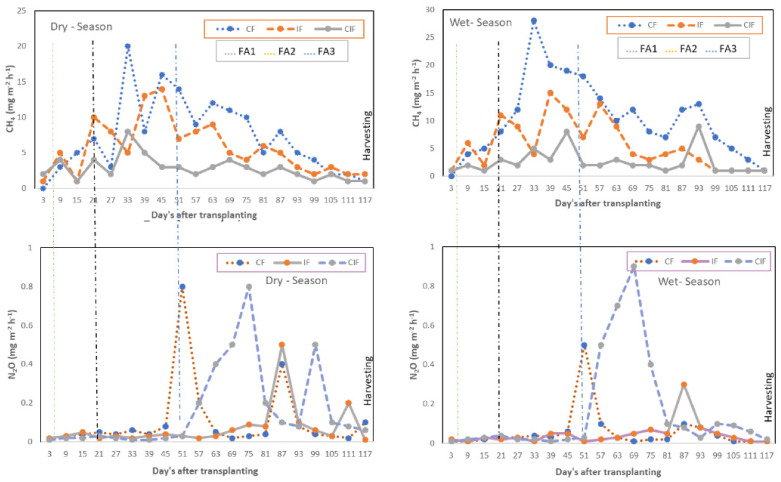
The seasonal emission of GHGs from the proposed rice fields by irrigation (CF, IF and CIF) and fertilizer treatments (FA-1, FA-2 and FA-3).

**Figure 9 sensors-22-04141-f009:**
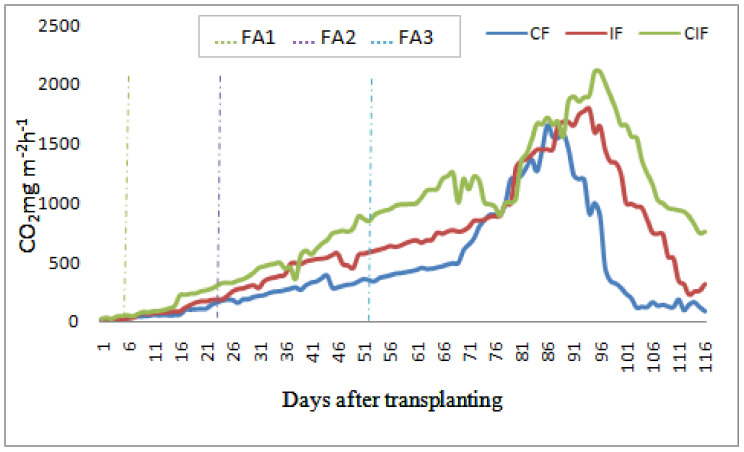
CO_2_ emission from rice fields (dry season).

**Table 1 sensors-22-04141-t001:** Various rice field management practices followed in both seasons.

Practice	Wet Season	Dry Season
Period of cropping	October 2020 to January 2021	January 2021 to May 2021
Crop duration	110 days	110 days
Pre-emergence herbicides	50 kg of dry sand + Butachlor 1.25 kg/ha	50 kg of dry sand + Butachlor 1.25 kg/ha
Post emergence herbicides	Pretilachlor + safener (Sofit) at 0.45 kg/ha	Pretilachlor + safener (Sofit) at 0.45 kg/ha
Fertilizer Application (FA)	120 kg/ha-DAP + 60 kg/ha-Urea (FA-1)	120 kg/ha-DAP + 60 kg/ha-Urea
	60 kg/ha-Urea (FA-2)	60 kg/ha-Urea
	120 kg/ha-P_2_O_5_ + 60 kg/ha-Urea (FA-3)	120 kg/ha-P_2_O_5_ + 60 kg/ha-Urea
Pesticide Application	Cartap hydrochloride 4% granules @ 18,750–25,000 g/ha propiconazole 25% EC @ 750//mLmLha	Cartap hydrochloride 4% granules @ 18,750–25,000 g/ha propiconazole 25% EC @ 750//mLmLha
Irrigation	CF–30 cm water (maintained) IF– 30 cm water (every 3 days) CIF–Soil moisture prediction and irrigation scheduling	CF–30 cm water (maintained) IF– 30 cm water (every 3 days) CIF–Soil moisture prediction and irrigation scheduling

**Table 2 sensors-22-04141-t002:** The estimated seasonal plant properties of proposed rice fields.

Irrigation Treatments	Avg. Plant Height (cm)		Avg. Panicle Length (cm)		Weight of 1000 Grains (gm)		Avg. Root Length (cm)		Avg. Root Weight (gm)	
	Wet	Dry	Wet	Dry	Wet	Dry	Wet	Dry	Wet	Dry
CF	101.9	104	30.1	32.4	25.1	29.5	23.5	24.3	17.5	20.2
IF	97.7	100	24.2	26.4	24.6	27.5	25.8	26.4	17.4	18.4
CIF	98.8	101	29.9	33.3	26.3	30.3	26.7	27.4	21.8	22.9

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
