# Peer review of "Sensing and Analysis of Greenhouse Gas Emissions from Rice Fields to the Near Field Atmosphere"

_sensors, 2022, doi:10.3390/s22114141_

Round 1
Reviewer 1 Report
The paper is well written and quite clear. The points for corrections are as follows:
Line 33: Please correct the numer of countries in Asia – there are no 114 but much less.
36-37: Why there are differences in GWP of various crops – more explanation is needed here.
53: The equation in Fig. 1 is not balanced - please correct.
151: Why the measurements did not last a whole year (365 days) and some months have been missing? Please explain.
Please rearrangee the scale in Fig. 7 (make the Y axis of the same scale). More descrition of the results shown in Fig. 7 is also needed in the manuscript text. Correct.
Rewrite the conclusions - please comment the values you have obtained. I would also recommend to put the conclusions as points (1......, 2......, etc.)
Author Response
Respected Sir/Madam
Please see the attachment

Reviewer 2 Report
This paper proposed a low-cost sensor-based greenhouse gases sampling and estimation system for rice fields. The proposed system performed well in both crop seasons and found that the adaptation of controlled intermittent flooding treatment into the rice fields would significantly reduce the greenhouse gas emission and increase rice productivity. This work also suggested some mitigation strategies that would reduce the production of greenhouse gas from rice fields.
This paper describes original research that I consider very interesting and timely. I think the research topic is appropriate for SENSORS (mdpi). The manuscript is well structured and well written, it is easy to read it. I believe the paper can be published after minor revision.
Comments
1. It will be interesting and helpful to find in the introductory sections some quantitative comparison between several methods available (micrometeorological and chamber) to measure greenhouse gas emissions from soil surface to atmosphere. Performance, uncertainty, …
2. I believe that the conclusions section could be improved with some additional quantitative details of the results.
3. What about future work and applications?
4. Have you a viable strategy to get a commercially available device in the near future?
Author Response

(The authors gave the same response as above.)

Reviewer 3 Report
To be honest, I'm quite interested in this study because the author proposes a new approach for monitoring GHG emissions with a low-cost sensor. However, there are a few basic problems about the measuring method proposed in this work that the author should address.
- When comparing the cost of technique that the author propose to the reference grade, it is clear that it is a long way off. Low-cost sensors, on the other hand, have fundamental flaws, such as data accuracy. When dealing with meteorological conditions and various gas concentrations, low-cost sensors have issues. Especially if you're working with a metal oxide sensor. How do you tackle these fundamental issues, as well as calibrate and validate the data that the low-cost sensor provides?
Please provide block diagram or block process to explain the three process were utilized to quantify GHGs fluxes in this research- How to calculate signal from the sensor into concentration, Please put in the manuscript
- Any formula to calculate of fluxes for each concentration which provide by sensor? Please put the formula in manuscript.
- Please clarify that the findings in the graphic for both CH4 and NO2 emissions are daily average results that are then converted to flux form.
- where is the CO2 result? could you explain it?
- Authors could put a highlight on the plot to show the FA-1, FA-2, and FA-3 procedures. Its aim was to make things clearer for readers.
- The authors also can give highlight in the plot about the harvesting period.
- The author indicates that meteorological data was recorded in the article; is there any explanation for the results produced by comparing the meteorological data that was recorded?
- When compared to N2O, CH4 is more magnitude occurred as shown in plot. Is there a different analysis for the fact that CH4 is more has magnitude occurred than N2O? Does this have anything to do with sensor capability, or is it more about soil chemical processes? Is there a link between meteorological factors and the consequence?
Author Response

(The authors gave the same response as above.)

Round 2
Reviewer 3 Report
Thank you to the authors for responding to the comments and revising the manuscript. However, several things in this manuscript need to be explained.
- In response to question 1, the author states that "since the estimation of GHG emission from rice field is not needed in finer scales (accuracy) we did not concentrated on accuracy." Could you please provide me with proof of your claim by providing a reference? Your assertion implies that measurements with a reference-grade analyzer are unnecessary. The precision of raw data is not necessary for the calculation of GHG emissions. Despite the fact that measurements are done by separating meteorological elements, I believe that low-cost sensors require specific consideration, including cross-sensitivity between the concentrations measured. For example, if you use the TGS2661 sensor type, this sensor indirectly measures ethanol, hydrogen, and isobutane, as stated in the datasheet. How can you be certain that the gas you're monitoring is solely methane? Even though you mention that it is not necessary to pay attention to the accuracy of raw data when measuring gas emissions, it is nevertheless employed in the calculation process. Where the accuracy of raw data must be addressed indirectly in the calculating process. So again, How do you validate or at least calibrate the sensor you are using to get the correct measurement data
- Is the vo symbol to represent the raw data value collected from the sensor in the calculation for computing GHG emissions? If this is the case, determine if the ppm concentration value can be utilized right away or if it must be processed beforehand. Please make any necessary clarifications and write them in the manuscript. Please make sure that each equation in the manuscript is numbered.
- Figure 9, where the findings of CO2 have a magnitude that suddenly declines twice, especially in the case of CF, is interesting. Please describe the events that occurred during that time period.
- Please can't simply provide a single line demonstrating how you highlight FA-1, FA-2, and FA-3. You can do this by using a different color for identifying and describing the color difference in the figure caption. Please, do it also for Figure 9.
Author Response
Respected Sir / Madam,
Please see the attachment
